# Role of Hypoxia in the Control of the Cell Cycle

**DOI:** 10.3390/ijms22094874

**Published:** 2021-05-05

**Authors:** Jimena Druker, James W. Wilson, Fraser Child, Dilem Shakir, Temitope Fasanya, Sonia Rocha

**Affiliations:** 1Centre for Gene Regulation and Expression, School of Life Sciences, University of Dundee, Dundee DD1 5EH, UK; j.druker@dundee.ac.uk; 2Department of Molecular Physiology and Cell Signalling, Institute of Systems, Molecular and Integrative Biology, University of Liverpool, Liverpool L69 7ZB, UK; James.Wilson3@liverpool.ac.uk (J.W.W.); F.Child@liverpool.ac.uk (F.C.); Dilem.Shakir@liverpool.ac.uk (D.S.); Temitope.Fasanya@liverpool.ac.uk (T.F.)

**Keywords:** hypoxia, cell cycle, HIF, PHDs, 2-OGDs, mitosis

## Abstract

The cell cycle is an important cellular process whereby the cell attempts to replicate its genome in an error-free manner. As such, mechanisms must exist for the cell cycle to respond to stress signals such as those elicited by hypoxia or reduced oxygen availability. This review focuses on the role of transcriptional and post-transcriptional mechanisms initiated in hypoxia that interface with cell cycle control. In addition, we discuss how the cell cycle can alter the hypoxia response. Overall, the cellular response to hypoxia and the cell cycle are linked through a variety of mechanisms, allowing cells to respond to hypoxia in a manner that ensures survival and minimal errors throughout cell division.

## 1. Introduction

The cell cycle is a process through which cells faithfully replicate their genetic material. Strict control over its progression is therefore needed to avoid errors that could result in cell death or cell malignancy. As such, understanding how the cell cycle is affected by external and internal stresses is of the utmost importance, in particular, the stress caused by hypoxia, or reduced oxygen availability. Hypoxia is an important factor in embryo development, but is also present in numerous pathological settings such as ischaemic events and cancer [1]. Oxygen is fundamental for both energy homeostasis and cellular viability, therefore, to deal with such stresses, cells possess complex response mechanisms that aim at restoring oxygen homeostasis. In this review, we highlight how hypoxia sensing and signalling pathways interface with the cell cycle, in addition to how the cell cycle affects the hypoxia sensing and response components in mammalian cells.

## 2. Hypoxia Signalling Pathway

In cells, oxygen is sensed by a class of 2-oxoglutarate, iron-dependent dioxygenases (2-OGD); the most recognized of which are the prolyl-hydroxylases (PHDs). These are instrumental in the signalling cascade initiated in response to a reduction in oxygen, which ultimately results in the activation of the hypoxia-inducible factor (HIF) transcription factor family. HIFs are heterodimers corresponding to a HIF-α (alpha) and HIF-1β (beta) subunit [2], while the PHDs present three isoforms (PHD1, PHD2, and PHD3) and possess a low affinity for oxygen so that any small deviation from normal oxygen concentrations will result in their enzymatic inactivation [3]. Under normal oxygen concentrations, the PHDs hydroxylate specific proline residues in the oxygen-dependent degradation domain (ODD) of HIF-α. This post-translational modification increases the binding affinity of the von Hippel-Lindau (VHL) protein, which acts as part of the recognition complex for the E3 ligase comprising of Elongin B/C, Cullin2, and RBX1 [4]. This complex ubiquitinates HIF-α, which marks the protein for proteasomal degradation. In the absence of either oxygen, iron (Fe^2+^), or 2-oxoglutarate (2-OG), the PHDs cannot function and the ability of VHL to promote ubiquitination of HIF is reduced, which results in the stabilization of HIF-α. Upon stabilization, and in combination with its binding partner HIF-1β, a HIF-1α-β heterodimer promotes the transactivation of a variety of target genes in response to reduced oxygen concentrations (for a recent review on HIF-dependent genes see [5]). Notably, it is important to highlight that, in addition to HIF, further transcription factors are also activated by hypoxia. These include Myc, p53, AP-1, Sp1, and NF-κB [6] (Figure 1).

Additional important 2-OGD enzymes, with links to hypoxia sensing and response, are JmjC-histone demethylases and TET DNA demethylases. Moreover, other JmjC-hydroxylases and RNA demethylases such as FTO and ALKBH5 are likewise potential contributors to the hypoxia response. However, additional work is needed to identify their true involvement. Interestingly, all of these aforementioned enzymes also have the potential to either directly or indirectly interface with the cell cycle (Figure 1).

## 3. Cell Cycle Overview

The cell cycle can be divided into two main stages: interphase and cell division (Figure 2). Interphase is where cells spend the majority of the time and can be subdivided into Gap1 (G1), S (DNA synthesis), and Gap2 (G2) phases; whereby following the G2-phase the cells enter mitosis (M) (cell division). Here, the duplicated DNA is distributed equally into two daughter cells, which then enter a new cell cycle (Figure 2). In G1, individual cells grow and their cellular content is duplicated, while in S-phase, the DNA is replicated, then in G2, the cell undergoes further growth and prepares for cellular division [7]. Finally, in mitosis, the cell divides its contents into two equal daughter cells. Here, depending on the cell type, cells will either enter another G1 stage or enter quiescence (also known as G0), whereby the cell exits the cell cycle and stops dividing [8].

The progression through the cell cycle is regulated by serine/threonine protein kinases known as cyclin-dependent kinases (CDKs), alongside their partner proteins, cyclins (Figure 2). However, CDK-independent roles have also been described for the cyclins in the regulation of apoptosis, differentiation, DNA repair, and metabolism reviewed in [9,10]. CDK-cyclin complexes phosphorylate specific substrates in each phase of the cell cycle, driving the transition to the next phase. The activation of the CDK depends on the association with the corresponding regulatory cyclin; therefore, the levels of each cyclin fluctuate during the cell cycle. This oscillation of the cyclin levels is regulated through their targeted degradation by different E3 ligase complexes in response to signalling pathways such as growth factor and oncogene activation [11]. CDK activity is also regulated by phosphorylation and by association with CDK-cyclin inhibitors such as p21 and p27, while the dysregulation of CDK activity is a common feature in many cancers [12,13].

Cell cycle progression is therefore a tightly regulated process that includes several checkpoints that are imperative during cell division to ensure the correct segregation of the genetic material between daughter cells. Furthermore, as the cell cycle is also a highly demanding and energy-consuming process, it follows that the oxygen-sensing system can directly impact cell cycle progression [14]. Indeed, key players in the hypoxia response are involved in the regulation of essential components during each phase of the cell cycle, at each of the transcriptional and protein stability levels, alongside localization and/or activity. Hence, oxygen deprivation or hypoxia must impact normal cell cycle progression [14]. In this review, we provide examples of how cell cycle progression in normal and cancer cells can be affected in response to oxygen levels, focusing on the molecular interplay between hypoxia and known cell cycle regulators.

## 4. Hypoxia Mediated Transcriptional Effects on Cell Cycle Components

Hypoxia plays a central role in diverse aspects of cancer biology as a hypoxic microenvironment is prevalent in solid tumours. It is known that oxygen levels regulate cell proliferation and that depending on the cell type, hypoxia can inhibit cell proliferation by inducing cell cycle arrest [15,16,17]. However, tumour cells often adapt to survive in such hypoxic conditions. This adaptation is partially promoted by the essential role HIF plays in the transcriptional regulation of genes associated with angiogenesis, apoptosis, cell proliferation, and energy metabolism [2]. HIF-dependent activation of these genes provides tumour cells with an adaptive advantage over normal cells during periods of hypoxic stress.

The Myc transcription factor, which is also associated with cell proliferation, is similarly overexpressed in various cancer types [18]. Myc is known to repress the expression of the CDK cyclin inhibitors p21 and p27 [18]. In hypoxia, the expression of p21 and p27 is induced in a HIF-1α dependent manner, partially by displacing Myc from their promoters [15]. Interestingly, HIF-2α cooperates with Myc and promotes cell proliferation in renal clear cell carcinomas (RCC) and other cell types [19]. In RCC, HIF-2α is also known to induce Cyclin D1 via its binding to an enhancer site [20] (Figure 3). The effects of HIF-2α on cell cycle control in RCC cells have become even clearer following data derived from treatments with a specific HIF-2α inhibitor (PT2399), in which the cell cycle was identified as the main signature controlled by HIF-2α [21,22].

The mitotic kinase—Aurora A (AURKA)—also has an important role in mitotic progression and has been related to oncogenic phenotypes [23]. AURKA is overexpressed in a variety of different tumours and has been implicated in cell transformation and centrosome amplification [24,25]. In hepatocellular carcinomas (HCC), the increased levels of AURKA do not always correlate with gene amplification [26]. Interestingly, AURKA was shown to be induced by hypoxia in a HIF-1α dependent manner in HepG2 cells [27] as HIF-1α binds to the AURKA promoter and upregulates its transcription, leading to an increase in cell proliferation [27]. This study proposed that AURKA might be involved in the hypoxia-induced proliferation of HCC tumours [27] (Figure 3). In this tumour type, hypoxia induction of HIF-1α and AURKA might be involved in promoting HCC proliferation [27]. However, in breast cancer cells (MCF-7, MDA-MB-231, and SK-Br3), microarray analysis in response to hypoxia illustrates that hypoxia downregulates AURKA [28]. In this study, the authors demonstrated that hypoxia downregulated AURKA via a HIF-1α dependent mechanism, suggesting that HIF-1α is a negative regulator of AURKA in breast cancer tumours [28]. These studies highlight the cell-specific nature of HIF’s function in cell cycle regulation.

Cell division cycle-associated protein 2 (CDCA2) has similarly been shown to be a target of the hypoxia pathway [29]. Analyses of publicly available ChIP-Seq and RNA-Seq datasets demonstrate that CDCA2 is induced in hypoxia and is important for the control of proliferation in prostate cancer cells. The intracellular signal transducer protein, SMAD3, binds to the CDCA2 promoter and recruits HIF-1α. HIF-1α/SMAD3 then mediates the transcriptional regulation of CDCA2 in hypoxic tumours, leading to cancer cell proliferation [29].

HIF’s control of the cell cycle is also mediated by its ability to induce a variety of microRNAs and long non-coding RNAs; this aspect was reviewed in [14,30].

Among all known hypoxia-stimulated transcription factors, the importance of NF-κB has been identified in a wide range of signalling systems, alongside its association with many different diseases. In contrast to its well-established role in immune and inflammatory responses, NF-κB activity is also involved in apoptosis, carcinogenic transformation, and cell cycle transition [31]. The NF-κB family consists of five distinct members, which include RelA (p65), RelB, c-Rel, NF-κB1 (p105/p50), and NF-κB2 (p100/p52), all of which share a conserved Rel homology domain [31]. Hypoxia activation of NF-κB occurs via several mechanisms, although some of the details still require further investigation. Hypoxia has been shown to control NF-κB via the action of PHDs and FIH [32,33,34], and requires the involvement of calcium/calmodulin-dependent kinase II (CAMKII), transforming growth factor kinase 1 (TAK1) and IκB kinase (IKK) [35,36,37].

NF-κB has several target genes identified with a role in cell cycle progression (reviewed in detail [31]). Ultimately, cyclin D1 provides a predominant link between NF-κB and cell cycle progression (Figure 3). Cyclin D1, in association with CDK4 and CDK6, promotes G1/S phase transition through CDK-dependent phosphorylation of retinoblastoma protein (pRB) [38]. This phosphorylation event releases the transcription factor E2F, which is required for the activation of S-phase-specific genes, although several studies have conversely reported that NF-κB can itself induce cell cycle arrest [39]. Overexpression of RelA has been shown to arrest cells at G1/S-phase transition [40]. c-Rel overexpression leads to cell cycle arrest through p53 protein stabilization, an important upstream activator of the CDK-inhibitor, p21 [41]. Additionally, p21 expression can be further increased by the Formin-2 (FMN2) protein, which is a component in the p14ARF tumour suppressor pathway [42]. In this study, FMN2 was shown to increase with hypoxia stimulation via an NF-κB-dependent mechanism [42]. Moreover, several additional studies have shown that NF-κB can be activated by a hypoxic environment [35,36,43,44].

Finally, a physical and functional interaction exists between the IKK/NF-κB signalling pathway and the cell cycle regulatory proteins of the E2F family that controls S-phase entry, which suggests that NF-κB plays a functional role in controlling cell division [45,46]. Indeed, direct phosphorylation of E2F by IKKα and IKKβ resulted in the nuclear accumulation and enhanced DNA binding of the E2F4/p130 repressor complex, which led to the suppression of E2F-responsive gene expression [45]. In contrast, E2F1 and NF-κB interactions were shown to control the timing of cell proliferation [46] (Figure 3). Despite all of these links between NF-κB and cell cycle regulation, a more systematic and mechanistic analysis of how NF-κB controls the cell cycle in hypoxia is needed.

## 5. Hypoxia Transcriptional-Independent Effects on Cell Cycle Components

Apart from transcription, hypoxia can alter a variety of other processes such as translation and post-translational modifications. As such, non-transcriptional mechanisms are also involved in hypoxia-induced cell cycle alterations (Figure 4).

### 5.1. Non-Transcriptional Role for HIF-1α

A non-transcriptional mechanism through which cell cycle progression is inhibited involves HIF-1α regulation of DNA helicase loading. During G1, under normal conditions, the MCM proteins assemble into a hexamer, poised in an inactive yet loaded state, by both Cdc6 and Cdt1 [47]. When transitioning from G1 into the S-phase, CDK2 catalyses the phosphorylation of the complex and promotes the nuclear export of the inhibitory Cdc6, therefore allowing for Cdc7-mediated phosphorylation of the MCM complex and the subsequent activation of DNA replication [48,49]. Upon hypoxic stress, HIF-1α interacts with Cdc6 to promote nuclear localization with the MCM complex [50]; while HIF-1α-bound Cdc6 increases MCM association with chromatin, the presence of HIF-1α prevents Cdc7 from phosphorylating the complex [51]. Consequently, replication origin firing is blocked as Cdc45 and DNA polymerase α cannot be recruited. Subsequently, HIF-dependent blocking of DNA replication and proliferation was observed across a variety of cell types including, cancer cell lines, haematopoietic stem cells, fibroblasts, and lymphocytes [51,52,53]. While the regulatory binding of HIF-1α to protein complexes inhibits the proliferative signals in some tissues, several cancer cells possess the capacity to proliferate even under hypoxic conditions [54].

### 5.2. HIF-Independent PHD-Dependent Roles in Cell Cycle

Given that oxygen is sensed by the 2-OGDs, it is likely that other aspects of the cell cycle could be regulated independently of HIF activity. This is definitely the case for the PHDs, whose association to cell cycle control has been best documented [14]. Of note, in addition to requiring oxygen as a co-substrate, their functional catalytic activity further requires the presence of additional co-factors: 2-oxoglutarate and iron [55,56]. PHDs can additionally monitor amino acid concentrations [57], cellular nutrients [58], and act as metabolic sensors to detect defects in metabolic pathways such as the tricarboxylic acid (TCA) cycle [59]. As such, PHD activity is similarly inhibited following an accumulation of metabolite intermediates due to mutations in the metabolic enzymes: fumarate hydratase (FH), succinate dehydrogenase (SDH), and isocitrate dehydrogenase 1 and 2 (IDH1 and IDH2) [60].

Whilst still controversial within the hypoxia field [61,62], over the last decade, emerging studies have highlighted alternative roles for PHDs, identifying substrates, in addition to HIF-α, which are involved in a myriad of different pathways and regulated through PHD-hydroxylation [3].

Interestingly, within these emerging substrates, both PHD1 and PHD3 have been shown to play critical roles in response to oxygen depletion: first, in the regulation of proteins involved in the DNA damage response, and second, in centrosome biogenesis [63,64]. These two processes are critical in regulating the cell cycle. In cultured human cells, Moser et al. (2013) showed that variations in oxygen concentration altered the mitotic cell cycle progression through PHD1-mediated hydroxylation of the centrosomal protein 192 (Cep192) [63]. Cep192 acts as a large scaffold protein in human centrosomes and its expression is required for centrosomal duplication and maturation in addition to centriole duplication [65,66,67]. The authors discovered that after two and four hours in 1% O_2_, the protein levels of Cep192 stabilised to those comparable to HIF-1α in U2OS cells. Cep192 undergoes a post-translational modification, whereby in normoxia, PHD1 hydroxylates proline 1717, which promotes Cep192 proteasomal degradation. However, unlike HIF-α, which is degraded through the VHL pathway, PHD1-mediated hydroxylation of Cep192 stimulates the recruitment of the SCF(Skp2) E3-ligase complex, which mediates the ubiquitination that promotes Cep192 proteasomal degradation. The study by Moser et al. (2013) ultimately provided a mechanistic link between PHD activity and the direct regulation of the cell cycle, highlighting the role that oxygen availability plays on the processes of centrosome duplication and maturation, in addition to centriole duplication through the modulation of Cep192 stability.

Alongside Cep192, PHD1 has likewise been shown to modulate protein levels of the G1 cyclin—cyclin D1 [68]. Targeted-hydroxylation of the Forkhead Box O3A (FOXO3A) transcription factor at proline residues 426 and 437 in normoxia was revealed to destabilise its interaction with the deubiquitinase USP9x and promote FOXO3a degradation, a process that leads to an accumulation of cyclin D1 [68]. Moreover, PHD1 regulation of cyclin D1 has further been demonstrated in breast cancer cells [69], whereupon inactivation of the *Egln2* (*Phd1*) gene, cyclin D1 levels, and mammary gland cell proliferation decreased.

Interestingly, it is not solely PHD1 that has been found to be involved in the regulation of cell cycle components. However, while PHD2 represents the most abundant PHD isoform, it has currently not been shown to interact with any validated cell cycle machinery components. However, as the predominant regulator in the hypoxia pathway, many studies have focused on the role PHD2 plays in various tumour types. Percy et al. (2006) proposed that PHD2 functions as a tumour suppressor [70]. Tao et al. (2016) revealed that PHD2 inhibited cell proliferation due to cell cycle arrest at the G1/S-phase transition [71]. Additionally, the authors demonstrated that the arrest in the cell cycle was facilitated by negative regulation of cyclin D1 in a PHD2 hydroxylation-dependent manner [71]. These findings correlate with previously published data indicating that PHD2 can act in a tumour suppressor role in cancer cells [72,73,74]. However, several other studies have shown that downregulation of PHD2 in tumour cells corresponds to a reduction in tumour growth [75,76]. This indicates that PHD2 can act as a tumour promoter or tumour suppressor, depending on cell and/or tissue type.

PHD3 has similarly been suggested to function as a tumour suppressor, whereby its expression induced apoptosis [77] and inhibited angiogenesis and tumour growth [78]. Equally, PHD3 is required to induce apoptosis and inhibit tumour growth in vivo [79]. Moreover, cell cycle distribution analysis revealed that the overexpression of PHD3, in conjunction with radiation in pancreatic cancer cells, results in a reduced S-phase and a lengthened G2/M phase [80].

p53 has an important role in cancer biology as an important regulator of the cell cycle and it is also activated in hypoxia [81]. PHD3 regulates p53 stability by hydroxylation at proline 359 [82]. This modification increases p53 association with USP7 and USP10 deubiquitinases, leading to reduced ubiquitination and enhanced stabilisation [82].

### 5.3. Hydroxylation and Phosphorylation Interplay

Proline hydroxylation usually acts as a docking site for protein–protein interactions, best exemplified by the HIF-1α/VHL association [83]. As such, it is possible that apart from ubiquitin ligases, proline hydroxylation could act as a binding site for other proteins including kinases or phosphatases (Figure 5). The AKT kinase is a key player in oncogenesis [84]. Guo et al. (2016) demonstrated that PHD2 mediated hydroxylation of AKT promotes the binding of this kinase to the VHL ubiquitin ligase. Interestingly, this interaction does not target AKT for degradation, but instead leads to AKT inactivation through PP2A phosphatase-mediated dephosphorylation. It was suggested that in hypoxic microenvironments or in cells lacking a functional VHL protein, AKT is hyper-activated, which promotes cancer cell survival [85]. These examples demonstrate that hydroxylation mediated by PHDs not only targets proteins directly for degradation, but also plays an important role in regulating protein–protein interactions.

Another such example suggests an interplay between two protein modifications as proline hydroxylation and phosphorylation have been recently reported to play a role in cell cycle progression [86]. In a study of the DYRK1 kinase, a conserved proline located in the kinase domain was shown to be hydroxylated by PHD1 [86]. Autophosphorylation in a specific tyrosine residue is critical for the enzymatic activity of DYRK1. Interestingly, PHD1 mediated hydroxylation of DYRK1 takes place during translation, but before tyrosine phosphorylation occurs. Furthermore, proline hydroxylation is important for the tumour suppression activity of DYRK1 toward the regulation of VHL E3 ligase activity and hence, the stability of VHL substrates critical for cell cycle progression such as HIF-2α, AURKA, and cyclin D1. In conclusion, Lee et al. (2020) showed that DYRK1 hydroxylation precedes the phosphorylation/activation of the kinase, which is required for VHL tumour suppression function [86,87]. Since PHD1 hydroxylates the proline within a conserved kinase domain (CMGC), the authors suggest that proline hydroxylation could be an essential mechanism in the catalytic activation of all eukaryotic CMGC kinases [86]. CMGC kinases have crucial roles in controlling the cell cycle and include CDKs, MAPK, and CDK-like kinases [88]. Unbiased proteomic studies are therefore needed to investigate the possibility that proline hydroxylation has a global effect on protein phosphorylation.

## 6. How the Cell Cycle Controls the Hypoxia Response

Since the cell cycle is a highly coordinated process and considering the published links to the regulatory roles of both HIF and the PHDs within the cell cycle, it could be speculated that either HIF and/or the PHDs would themselves be subject to regulatory control by the cell cycle. Indeed, evidence of such control already exists.

### 6.1. Cell Cycle Components-Mediated Control of HIFs

In addition to its regulatory dioxygenases, several kinases directly involved in the cell cycle have similarly been shown to directly phosphorylate and regulate HIF-α [89].

Indeed, CDK1 has been shown to directly phosphorylate HIF-1α and promote HIF stabilization and activity [90]. Additionally, CDK2 promotes HIF-1α transactivation domain function, alongside upregulating HIF-1α-mediated expression of downstream target genes [91]. However, the coupling of HIF-1α to CDK2 presents a dualistic function, whereby CDK2-cyclin E stimulates HIF-1α degradation through chaperone-mediated autophagy during the G1/S-phase transition. This, in turn, circumvents the inhibitory effects on DNA replication elicited through any HIF-1α-MCM interaction. In contrast, CDK1-cyclin B complexes block the lysosomal degradation pathway, a process that promotes HIF-1α protein stabilisation and target gene activation [91]. Other important cell cycle regulators such as Aurora B and PLK3 have been shown to directly phosphorylate HIF-1α [92,93]. While Aurora B promoted ChIP-dependent degradation of HIF-1α [92], PLK3 phosphorylation is important for the reduced half-life of HIF-1α by an unidentified mechanism [93].

### 6.2. Cell Cycle Components-Mediated Control of PHDs

PHD1 activity is similarly the subject of cell cycle regulation by the cyclin-dependent kinases [94]. Ortmann and co-authors (2016) discovered that the serine 130 residue of PHD1 is phosphorylated, but that phosphorylation was dependent upon CDK activity, cell cycle stage, and specific oncogenic signals. The authors explain that phosphorylation was regulated by the activity of the interphase CDKs (2, 4, and 6), but not CDK1, and by activation of the proto-oncogenes—Myc and E2F1—both known cell cycle regulators [95,96]. Moreover, whilst phosphorylation does not functionally alter PHD1 hydroxylase activity in vitro, it does govern PHD1 target specificity between HIF-1α and Cep192, whereby phosphorylation increased PHD1 activity toward Cep192 and reduced the interaction with HIF-1α. This study by Ortmann and co-authors, therefore, establishes a mechanistic link between the cell cycle and the regulation of PHD1 activity in cells. Cell cycle mediators such as CDKs can temporally control the activity of PHDs, whereby specific post-translational modifications (PTMs) can alter their target specificity and direct them to precise targets in response to cellular signals [94]. PHD1 has further been shown to undergo phosphorylation in response to JNK2 activation [97]. JNK2 is also a regulator of the cell cycle [98]. Phosphorylation of serine 74 and serine 162 was detected in breast cancer cells by mass spectrometry. This phosphorylation was associated with increased PHD1 activity toward HIF-1α. However, the kinase controlling these modifications was not investigated.

PHD2 is similarly phosphorylated on serine 125, in normoxia, by the mitogen-activated serine/threonine kinase p70S6K, whereby phosphorylation enhances PHD2-mediated degradation of HIF-1α [99]. However, in colorectal cancer cells, the PP2A phosphatase and its regulatory subunit B55α dephosphorylate the serine 125 residue, which reduces PHD2 activity, thereby enhancing HIF-1α accumulation and ultimately stimulating autophagy-mediated cell survival [99].

More recently, the anaphase promoting complex (APC) activator CDC20 has been found to mediate the degradation of PHD3 in HCC cells, ultimately increasing the stability and activity of HIF-1α [100].

## 7. Conclusions

As the cell cycle is a process required to replicate the cell’s genome faithfully, therefore, it follows that mechanisms are in place to integrate stress signals with the cell cycle machinery (Table 1). In addition, stress pathways such as those induced by hypoxia that control cell cycle processes should also receive input from the components of the cell cycle. Additional studies of how 2-OGD enzymes interface with the cell cycle will expand our knowledge of this important crosstalk between hypoxia and cell cycle signalling pathways.

## Figures and Tables

**Figure 1 ijms-22-04874-f001:**
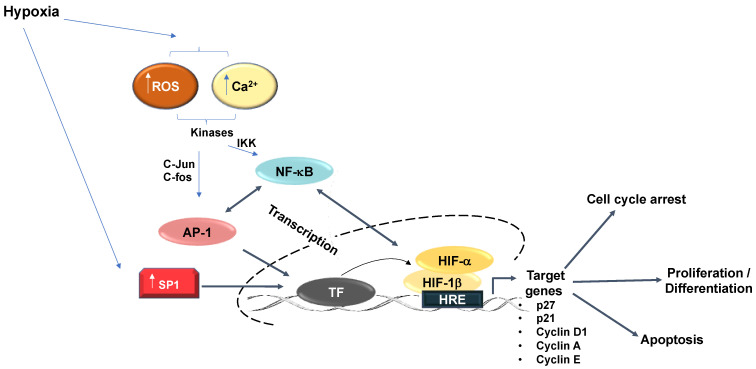
Hypoxia-relevant transcription factor crosstalk. Additional transcription factors than HIF regulate the hypoxia response. Hypoxia increases SP1 expression, which increases SP1 binding to gene promoters and upregulated transcriptional activation of downstream targets. Hypoxia further promotes increased SP1 binding to an upstream GC Box to augment the hypoxia-response element-dependent downstream gene activation. Hypoxia induces AP-1 activity. Mitogen-activated protein kinases (MAPK) activate AP-1 subunits (c-jun/c-fos) in response to hypoxia. Activation of AP-1 heterodimers predominantly exert their effects in cooperation with additional transcription factors including NFκB and HIF-1α to activate common target genes including regulation of cell proliferation and apoptosis. NFκB likewise regulates HIF-1α transcription.

**Figure 2 ijms-22-04874-f002:**
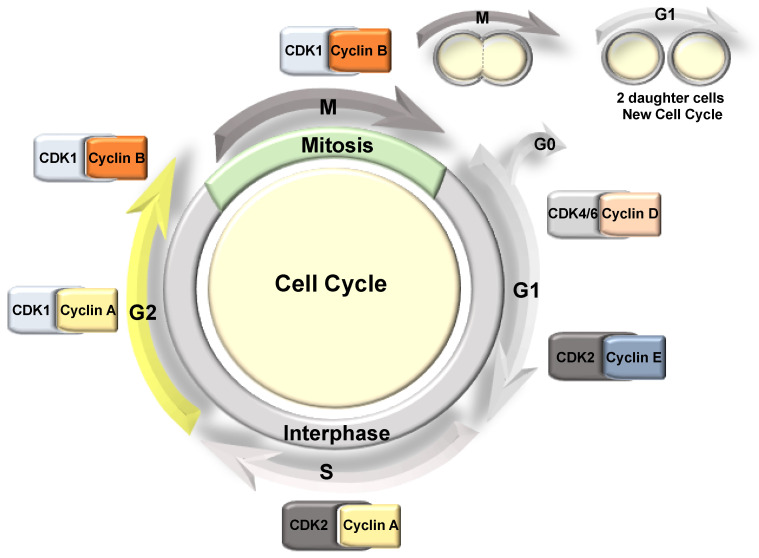
The mammalian cell cycle phases. The cell cycle is divided into two main phases: interphase that includes G1, S, and G2, and mitosis. Each of these stages is associated with a CDK/Cyclin pair, controlling the progression of the cell cycle.

**Figure 3 ijms-22-04874-f003:**
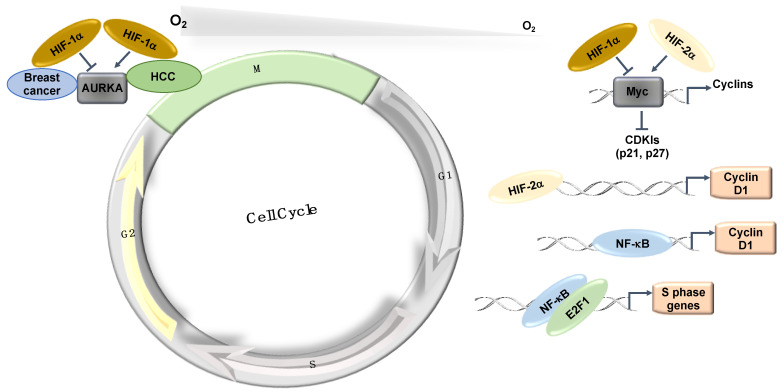
Hypoxia-induced transcriptional effects on the cell cycle. HIFs and NF-κB control the expression of a number of important components of the cell cycle machinery and control mechanisms. HCC: Hepatocellular carcinoma.

**Figure 4 ijms-22-04874-f004:**
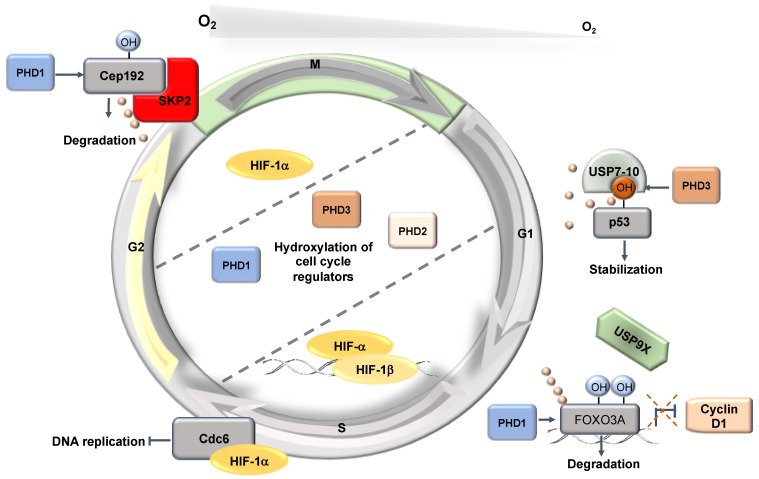
Hypoxia-induced post-transcriptional effects on the cell cycle. HIF has a non-transcriptional effect controlling DNA replication. In addition, PHD1 and PHD3 have non-HIF targets involved in the control of the cell cycle.

**Figure 5 ijms-22-04874-f005:**
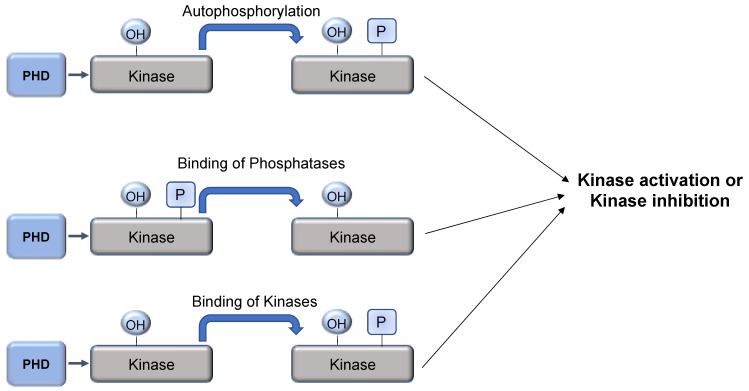
PHD-mediated hydroxylation as a mechanism to regulate kinase activity. Hydroxylation of kinases can lead to induction of autophosphorylation, recruitment of phosphatases, or even other kinases. This will result in changes to kinase activity.

**Table 1 ijms-22-04874-t001:** Summary of the interaction between hypoxia and the cell cycle.

Cell Cycle Control Mechanisms in Hypoxia
**Transcriptional effects**
Hypoxia induces the activation of several transcription factors: HIFs, Myc, p53, AP-1, SP1 and NFκB are all involved in the transcriptional regulation of cell cycle genes e.g., p27, p21, cyclin D1, A, E.
Myc regulates the expression of cyclins and CDKI. Performs the opposite role of the HIF-α subunits in Myc transcriptional regulation.
HIF-2α induces cyclin D1 expression.
HIF-1α can both induce or downregulate AURKA expression.
HIF-1α/SMAD3 regulates CDCA2 expression.
NFκB regulates cyclin D1 expression.
NFκB/E2F1 regulates S-phase genes expression.
**Transcription-independent effects**
HIF-1α interacts with Cdc6, which blocks DNA replication.
PHD1 mediated proline hydroxylation regulates Cep192 and FOXO3A stability as well as DYRK1 activity.
PHD3 mediated proline hydroxylation regulates p53 stability.
PHD2 mediated proline hydroxylation regulates AKT activity.
**Regulation of HIFs/PHDs by cell cycle components**
CDK1 phosphorylates HIF-1α promoting its stabilization and activity.
CDK2 regulates HIF-1α transactivation activity, while CDK2-cyclin E promotes HIF-1α degradation through autophagy.
Aurora B phosphorylates HIF-1α promoting its degradation.
PLK3 phosphorylates HIF-1α regulating its half-life.
CDK2, 4 and 6 phosphorylate PHD1 at serine 130 regulating its target specificity between HIF-1α and Cep192.
JNK2 regulates the phosphorylation of PHD1 serine 74.
p70S6K phosphorylates PHD2 at serine 125, while is dephosphorylated by the PP2A-B55α phosphatase. This phosphorylation regulates PHD2 activity.
CDC20 mediates PHD3 degradation, increasing HIF-1α stability and activity.

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
