# Peer review of "Role of Hypoxia in the Control of the Cell Cycle"

_ijms, 2021, doi:10.3390/ijms22094874_

Round 1

Reviewer 1 Report

The Authors presented an extensive review of role of hypoxia in control of cell cycle and focused on the role of transcriptional and post-transcriptional mechanisms induced by hypoxia which influence on cell cycle control. The review has defined the current knowledge in the field of hypoxia and cell cycle control pathways in all aspect presented in literature so far. All data were supported by highly selected references. The manuscript is well written, with precise description of each mechanism.

I have no major remarks, but a table of the cell cycle control mechanisms (transcriptional and transcription-independent effects) described in this article will be appropriate for summarizing the data and helping the reader organize the information.

Additionally, I strongly encourage the Authors to prepare a list of abbreviations that will be very helpful while reading the article.

The font on the graphs are disorderly. It is possible to change that? The elements describing the O2 concentrations on Figure 3 and 4 are hardly visible, please correct.

Is there any information about cell cycle regulation by hypoxia in lung cancer cells?

Minor remarks:

Line 128: hepatocellular carcinomas

Line 135: please add the breast cell lines names

Line 161: pRB, please describe

Line 201: 2-OGDc, please describe

Line 222: The sentence is not clear, please correct: firstly… and secondly…?, [59, 60]: ?

Line 228: human centrosomes

Line 336: PTMs, please describe

Line 366: the acknowledgements section should be filled or remove.

Line 502: reference [59] contains doi in an inappropriate format (website)

Author Response

The Authors presented an extensive review of role of hypoxia in control of cell cycle and focused on the role of transcriptional and post-transcriptional mechanisms induced by hypoxia which influence on cell cycle control. The review has defined the current knowledge in the field of hypoxia and cell cycle control pathways in all aspect presented in literature so far. All data were supported by highly selected references. The manuscript is well written, with precise description of each mechanism.

We thank the reviewer for taking the time to give our review feedback. We will answer the points raised below.

I have no major remarks, but a table of the cell cycle control mechanisms (transcriptional and transcription-independent effects) described in this article will be appropriate for summarizing the data and helping the reader organize the information.

Thank you for this suggestion, we have now compiled a table as suggested.

Additionally, I strongly encourage the Authors to prepare a list of abbreviations that will be very helpful while reading the article.

We have now included a list of abbreviations as suggested.

The font on the graphs are disorderly. It is possible to change that? The elements describing the O2 concentrations on Figure 3 and 4 are hardly visible, please correct.

We thank the reviewer for pointing this mistakes and we have now standardised the fonts and increased visibility.

Is there any information about cell cycle regulation by hypoxia in lung cancer cells?

There are indirect links in lung cancer, but none directly investigating cell cycle regulation by hypoxia in lung cancer.

Minor remarks:

Line 128: hepatocellular carcinomas

We made have this change as suggested

Line 135: please add the breast cell lines names

We have now included the names of the breast cancer cells used in the published study as requested.

Line 161: pRB, please describe

We apologise, we have now described this

Line 201: 2-OGDs, please describe

We describe these in line 34, as 2-oxoglutarate, iron-dependent dioxygenases

Line 222: The sentence is not clear, please correct: firstly… and secondly…?, [59, 60]: ?

We apologise for this error, we have now rephrased it to make this sentence clearer.

Line 228: human centrosomes

This has been corrected.

Line 336: PTMs, please describe

We have added a description

Line 366: the acknowledgements section should be filled or remove.

We have removed this section.

Line 502: reference [59] contains doi in an inappropriate format (website)

We have now corrected this.

Reviewer 2 Report

This is a review article describing the interplay between cell cycle regulation and hypoxic response pathways. Overview of HIF signaling and cell cycle regulation is described in the first part, followed by the description of HIF regulating cell cycle, and cell cycle-dependent regulation of HIF activity.

Although there are numbers of study on cell cycle regulation under hypoxic condition, comprehensive review of it is hardly found. It should serve as an useful article for researchers of broad range.

Following points should be properly modified.

  1. Figures contain letters with different fonts. It will make easier to read if one font type (such as Arial) is used throughout the figures.
  2. line 103: Authors should also refer to the normal cells (eg., MCB 2003, PMID 12482987), not just cancer cells.
  3. line 151 and below: Authors should describe more in detail how hypoxic condition affects NF-kB activity, rather than general explanation of this transcription factor.
  4. Figure 3: it is not clear how AURKA is regulated by HIF-1a. Information of cell type should be added.
  5. line 247: Phd1 is Egln2.
  6. line 254: This sentence appears to contain contradictory facts. Would PHD2 be called ‘tumor suppressor?’
  7. It will become easier to understand if a figure is added to section 5.3. (starting from line 272)

     8. Several typos needs to be corrected: eg., line 228 centrossomes, line  231 U20S

Author Response

Comments and Suggestions for Authors

This is a review article describing the interplay between cell cycle regulation and hypoxic response pathways. Overview of HIF signaling and cell cycle regulation is described in the first part, followed by the description of HIF regulating cell cycle, and cell cycle-dependent regulation of HIF activity.

Although there are numbers of study on cell cycle regulation under hypoxic condition, comprehensive review of it is hardly found. It should serve as an useful article for researchers of broad range.

We thank this reviewer for the positive comments and feedback, and have answered the points raised below.

Following points should be properly modified.

  1. Figures contain letters with different fonts. It will make easier to read if one font type (such as Arial) is used throughout the figures.

We apologise for this oversight. We have now standardised the font used as suggested.

  1. line 103: Authors should also refer to the normal cells (eg., MCB 2003, PMID 12482987), not just cancer cells.

We thank the reviewer for this suggestion and have changed the sentence to include this suggestion

  1. line 151 and below: Authors should describe more in detail how hypoxic condition affects NF-kB activity, rather than general explanation of this transcription factor.

Thank you for this suggestion, we have included a more detailed description of this important aspect as suggested.

  1. Figure 3: it is not clear how AURKA is regulated by HIF-1a. Information of cell type should be added.

We have added the cell type specificity on the regulation of AURKA by HIF-1alpha

  1. line 247: Phd1 is Egln2.

We have now corrected this mistake.

  1. line 254: This sentence appears to contain contradictory facts. Would PHD2 be called ‘tumor suppressor?’

We have clarified this sentence to make it easier to understand. PHD2 has cell type specific functions. In certain cancers it can acts as a tumour suppressor but not in others.

  1. It will become easier to understand if a figure is added to section 5.3. (starting from line 272)

We have created a new Figure to help detail this section as suggested.

  1. Several typos needs to be corrected: eg., line 228 centrossomes, line  231 U20S

We thank the reviewer for highlighting these errors, we have attempted to detect and correct all typos in the document.

Reviewer 3 Report

This review summarized the regulatory mechanisms in the control of cell cycle under hypoxia. It discussed the transcriptional and transcriptional-independent effects on cell cycle mediated by hypoxia. Also, the review demonstrated the reciprocal interactions between cell cycle controls and the hypoxia response. 

There are several minor concerns regarding this review.

  1. line 102, "in this section, we provide examples... ..." Is the"section" referring to section 4?
  2. line 136, "indicating that HIF-1a can act as a negative regulator... ..." This sentence is really confusing. Is the downregulation of AURKA due to HIF-1a or hypoxia?  It is not a precise way to summarize the cited study.
  3. line 142, "The intracellular signal ... ... leading to cancer cell proliferation" is too long. can be separated into two shorter sentences. similar long sentences such as line 159-161;line 164-167.etc are not concise.
  4. line169,"stimulation and via an NF-kB..." is a weird sentence.
  5. line 203, "Where the regulatory ... ..." I guess the author wants to say "whereas"?
  6. The character styles used in the figures are not consistent. I see at least two different styles.
    Overall the review is fine. However, it needs to go through editing to make it easy to read. 

Author Response

This review summarized the regulatory mechanisms in the control of cell cycle under hypoxia. It discussed the transcriptional and transcriptional-independent effects on cell cycle mediated by hypoxia. Also, the review demonstrated the reciprocal interactions between cell cycle controls and the hypoxia response. 

We thank the reviewer for taking the time to give us feedback on our review. We will answer the specific points raised below

There are several minor concerns regarding this review.

  1. line 102, "in this section, we provide examples... ..." Is the"section" referring to section 4?

We have made this clearer in the text. We meant to say ‘review’ instead of ‘section’, our apologies.

  1. line 136, "indicating that HIF-1a can act as a negative regulator... ..." This sentence is really confusing. Is the downregulation of AURKA due to HIF-1a or hypoxia?  It is not a precise way to summarize the cited study.

We apologise for the lack of clarity, we have now rephrased it to make it easier and more faithful to the original study as suggested. The authors demonstrate that in breast cancer cells, hypoxia reduces AURKA mRNA levels in a HIF-1alpha dependent manner.

  1. line 142, "The intracellular signal ... ... leading to cancer cell proliferation" is too long. can be separated into two shorter sentences. similar long sentences such as line 159-161;line 164-167.etc are not concise.

We thank the reviewer for these suggestions. We have now changed all of the sentences mentioned to shorter ones.

  1. line169,"stimulation and via an NF-kB..." is a weird sentence.

We apologise as this sentence has a typo that makes it strange. We have now rephrased it to make it clearer.

  1. line 203, "Where the regulatory ... ..." I guess the author wants to say "whereas"?

Indeed, we apologise for this, and have now corrected it.

  1. The character styles used in the figures are not consistent. I see at least two different styles.

We have now made this more consistent and used only one font type.

Overall the review is fine. However, it needs to go through editing to make it easy to read. 

We thank the reviewer for pointing out these errors in the text. We have attempted to correct all the typos we could detect and to make all the sentences clearer and easier to read.

Round 2

Reviewer 3 Report

My concerns are well addressed.

Author Response

thank you